# Aberrant Activation of the STING-TBK1 Pathway in γδ T Cells Regulates Immune Responses in Oral Lichen Planus

**DOI:** 10.3390/biomedicines11030955

**Published:** 2023-03-20

**Authors:** Shan Huang, Ya-Qin Tan, Gang Zhou

**Affiliations:** 1The State Key Laboratory Breeding Base of Basic Science of Stomatology (Hubei-MOST) & Key Laboratory of Oral Biomedicine Ministry of Education, School and Hospital of Stomatology, Wuhan University, Wuhan 430079, China; 2Department of Oral Medicine, School and Hospital of Stomatology, Wuhan University, Wuhan 430079, China

**Keywords:** oral lichen planus, γδ T cells, STING-TBK1 pathway, interferon-γ

## Abstract

Oral lichen planus (OLP) is a chronic T cell-mediated inflammatory disease. Interferon (IFN)-γ has been suggested to be vital for the OLP immune responses. A prominent innate-like lymphocyte subset, γδ T cells, span the innate–adaptive continuum and exert immune effector functions by producing a wide spectrum of cytokines, including IFN-γ. The involvement and mechanisms of γδ T cells in the pathogenesis of OLP remain obscure. The expression of γδ T cells in lesion tissues and in the peripheral blood of OLP patients was determined via flow cytometry and immunohistochemistry, respectively. Human leukocyte antigen-DR (HLA-DR), cluster of differentiation (CD) 69, Toll-like receptors (TLRs), natural killer group 2, member D (NKG2D) and IFN-γ were detected in γδ T cells of OLP patients using flow cytometry. Additionally, the involvement of stimulator of the interferon genes (STING)-TANK-binding kinase 1 (TBK1) pathway in γδ T cells was evaluated by multi-color immunofluorescence. Western blotting was employed to investigate the regulatory mechanisms of γδ T cells in OLP. γδ T cells were significantly upregulated in the lesion tissues, whereas their peripheral counterparts were downregulated in OLP patients. Meanwhile, increased frequencies of local CD69^+^ and NKG2D^+^ γδ T cells and peripheral HLA-DR^+^ and TLR4^+^ γδ T cells were detected in OLP. Furthermore, significant co-localization of STING and TBK1 was observed in the γδ T cells of OLP lesions. In addition, enhanced IFN-γ and interleukin (IL)-17A were positively associated with the activated STING-TBK1 pathway and γδ T cells in OLP. Taken together, the upregulated STING-TBK1 pathway in activated γδ T cells might participate in the regulation of immune responses in OLP.

## 1. Introduction

Oral lichen planus (OLP) is a chronic T cell-mediated inflammatory disease with unknown etiology, clinically characterized by bilateral symmetrical white lines or patches on oral mucosa [1,2]. Two distinctive clinical patterns of OLP are recognized as nonerosive (reticular and atrophic) and erosive forms [3]. Microscopically, dense subepithelial lymphocytic infiltrate and degeneration of basal keratinocytes are identified in OLP lesions [4]. Antigen presentation; T cell activation, proliferation and migration; and keratinocyte apoptosis are widely accepted to be involved in the pathogenesis of OLP [4]. IFN-γ, the only type II interferon and key T helper (Th) 1 lineage-specific cytokine, activates cytotoxic CD8^+^ T cells and maintains the expression of major histocompatibility class (MHC) on keratinocytes in the advanced stage of OLP [5,6]. Therefore, increased IFN-γ may underlie the Th1-biased immune responses in OLP [5,6].

γδ T cells expressing γδ T cell receptor (γδ TCR) are a prominent innate-like lymphocyte subset spanning the innate–adaptive continuum [7]. With the functional effects of γδ TCR, natural killer group 2, member D (NKG2D) and Toll-like receptors (TLRs), γδ T cells can recognize a wide range of antigens without MHC restriction and act as early sensors to respond to cellular stress or infection [7,8]. By secreting pro-inflammatory cytokines, including IFN-γ, IL-17 and tumor necrosis factor-α (TNF-α), γδ T cells play pivotal roles in initiating and regulating inflammatory responses in the mucosal barrier [5,9,10]. An accumulating mound of studies have revealed the involvement of γδ T cells in the pathogenesis of several autoimmune diseases, such as psoriasis, systemic lupus erythematosus (SLE) and rheumatoid arthritis (RA) [11,12,13]. Meanwhile, proliferation of naïve CD8^+^ T cells and their differentiation into cytotoxic T lymphocytes can be induced by γδ T cells [14]. Additionally, γδ T cells serve as a predominant innate source of IFN-γ, which is required for Th1-dominated immune responses [15,16]. Increased IFN-γ has been found in mononuclear cells throughout the subepithelial infiltrate and in T cells isolated from OLP lesions, which was correlated with disease severity [17,18,19]. In our previous study, upregulated peripheral IFN-γ was demonstrated to participate in the immunoregulatory mechanisms of OLP [20,21]. Whether γδ T cells contribute to the immune inflammatory condition of OLP remains obscure.

Stimulator of interferon genes (STING), a dimeric transmembrane protein, is mainly distributed in immune cells, orchestrating the autoimmune and inflammatory responses. After recognition and binding of free cytoplasmic DNA, activated cyclic guanosine monophosphate-adenosine monophosphate synthase (cGAS) synthesizes cyclic guanosine monophosphate–adenosine monophosphate (cGAMP). By binding to cGAMP, STING undergoes a translocation and further activates the downstream TANK-binding kinase 1 (TBK1), which eventually triggers production of IFNs [22,23]. Self-DNA released by apoptotic cells activates the STING-TBK1 pathway and then induces a variety of autoimmune inflammatory diseases, including RA and SLE [23,24]. Recently, the STING-TBK1 pathway has been suggested to induce the differentiation of T cells and enhance the T-cell-mediated IFN responses [24,25,26]. STING ligands could promote cytokine induction in tumor-reactive human γδ T cells [27]. However, the involvement of the STING-TBK1 pathway in γδ-T-cell-mediated immunity of OLP has been elucidated.

In the present study, the expression patterns of γδ T cells in the peripheral and local immune environment of OLP were investigated. Meanwhile, activation and antigen-presentation-associated molecules were detected on γδ T cells. The expression level of IFN-γ secreted by γδ T cells was also measured. Furthermore, the activation of the STING-TBK1 pathway, together with their co-localization with γδ T cells, were explored to identify the possible involvement of γδ T cells and the STING-TBK1 pathway in the pathogenesis of OLP.

## 2. Materials and Methods

### 2.1. Participants and Samples

Ethical approval was granted by the Ethical Committee of Wuhan University, and sample collection was in accordance with the principles of the Declaration of Helsinki. In total, 39 clinically and histopathologically confirmed OLP patients and 27 age-and-sex-matched healthy controls were enrolled with informed consent in this study. Oral mucosal tissue specimens and peripheral blood were obtained. The overall clinical data of participants are displayed in Table 1. Inclusion and exclusion criteria for OLP and healthy participants were in line with our previous study [20]. Only these presenting with symmetrical white lesions and typical histopathological characteristics of OLP were included, in accordance with modified WHO criteria [1]. These oral lichenoid lesions and OLP lesions with dysplasia were excluded. OLP participants were recruited strictly without any other visible oral lesions nor systemic disorders.

### 2.2. Samples Preparation for Flow Cytometry

All peripheral blood samples were disposed of within 48 h and were isolated by Ficoll-Paque density-gradient centrifugation for preparation of peripheral blood mononuclear cells (PBMCs). Then, the harvested PBMCs were washed twice before incubating with RBC lysis buffer (Beyotime, Shanghai, China, Catalog #C3702) for 3 min at room temperature. The cells were resuspended gently in PBS for further flow cytometry.

OLP-lesion tissues and healthy oral-mucosal biopsies were conserved in RPMI-1640 (Biological Industries, Kibbutz Beit Haemek, Israel, Catalog #BISH0266) supplemented with a 1% penicillin–streptomycin–gentamicin solution (Solarbio, Beijing, China, Catalog #P1410) at 4 °C within 3 h. The samples were minced with scissors into small pieces of approximately 2–3 mm in size after being washed in PBS containing a 2% penicillin–streptomycin–gentamicin solution, and then incubated with 80 μg/mL DNase I, RNase-free (Vazyme, Nanjing, China, Catalog #EN401-01), 200 μg/mL collagenase I (Biosharp, Hefei, China, Catalog #BS032A) and RPMI-1640 at 37 °C for 1 h. After incubation, the digestion reaction was stopped by adding the equal volumes of 2% fetal bovine serum (FBS). A 70 μm cell strainer was utilized to obtain dissociated cells. The isolated cells were then washed and treated with RBC lysis buffer for 3 min at room temperature to eliminate any erythrocytes contamination. After being washed and resuspended in RPMI-1640, the cells were activated using Cell Activation Cocktail (PMA/ionomycin/brefeldin A, Biolegend, San Diego, CA, USA, Catalog #423303) for 6 h at 37 °C. These activated cells were collected after incubation and prepared for further flow cytometry.

### 2.3. Multiparameter Flow Cytometry

For PBMCs isolated from venous blood, cells were stained by FITC-conjugated anti-CD3 (BioLegend, Catalog #300406) and PE-conjugated anti-TCRγδ (BioLegend, Catalog #331210) to gate CD3^+^ γδ T cells. In addition to the above-mentioned surface maker antibodies, PBMCs were simultaneously stained with APC-H7-conjugated anti-HLA-DR (BD Pharmingen™, San Jose, CA, USA, Catalog #561358), Percp 5.5-conjugated anti-TLR4 (BD OptiBuild™, San Jose, CA, USA, Catalog #745946) and APC-conjugated anti-NKG2D (BD Pharmingen™, San Jose, CA, USA, Catalog #558071) in PBS supplemented with 2% FBS for 20 min at room temperature in the dark.

Cells collected from tissue samples were labelled for surface antigens in the same buffer for 20 min at room temperature in the dark. The cells were stained with the following directly conjugated antibodies: anti-CD3 (FITC, BioLegend) in combination with anti-TCR γδ (PE, BioLegend) or anti-TCR αβ (PE, BioLegend, Catalog #306708), anti-CD69 (PE-Cy7, BioLegend, Catalog #310912), anti-HLA-DR (APC, BD Pharmingen™), anti-TLR4 (BV421, BioLegend, Catalog #312811) and anti-NKG2D (APC, BD Pharmingen™). Following surface-molecule staining, the cells were placed into 0.5 mL Fixation Buffer (Biolegend, Catalog #420801) for 20 min at room temperature.

To detect the expression of the pro-inflammatory cytokine IFN-γ in γδ T cells, Intracellular Staining Permeabilization Wash Buffer (Biolegend, Catalog #421002) was then used to wash and resuspended fixed cells. Afterwards, the cells labelled with surface markers’ antibodies in permeabilization wash buffer were incubated with BV650-conjugated anti-IFN-γ (BD Horizon™, San Jose, CA, USA, Catalog #563416) for 20 min at room temperature.

Appropriate isotype controls were used for all staining. All stained cells were resuspended in 0.5 mL PBS supplemented with 2% FBS and analyzed using CytoFLEX LX (Beckman Coulter, Brea, USA). A minimum of 10^6^ events was collected, and live cells were gated according to forward and side-scatter properties. Absolute cell count was determined as the percentage of cells and total cells. Flow cytometric data were prepared for presentation and analyzed via CytExpert 2.3 software (Beckman Coulter, Brea, CA, USA).

### 2.4. Immunohistochemistry Assay

Formalin-fixed, paraffin-embedded tissue sections were sectioned at a thickness of 4 μm, and then routinely deparaffinized in xylene and rehydrated in graded alcohol solutions. Antigen retrieval was performed with sodium citrate buffer (10 mM, PH = 6.0) using microwaves for 20 min. Prior to incubation with the primary antibodies, all the sections were treated with 3% H_2_O_2_ for 20 min at room temperature to block endogenous peroxidase activities. Sections were co-incubated with 5% BSA buffer for 1 h to attenuate non-specific protein binding. Then, slides were incubated with anti-γδ TCR (1:10, Thermo Fisher, Waltham, MA, Catalog #5A6.E9), anti-STING (1:200, Cell Signaling Technology, Danvers, MA, USA, Catalog #D2P2F) and anti-NAK/TBK1 (1:250, Abcam, Cambridge, UK, Catalog #EP611Y) primary antibodies in a moist chamber at 4 °C overnight, separately. Blank controls were treated with phosphate buffered saline (PBS) buffer instead. Then, HRP polymer-conjugated secondary antibodies were used for 1 h at 37 °C. The slides were visualized with diaminobenzidine (DAB) solution, followed by hematoxylin counterstaining. The positive staining was measured from at least 4 randomly selected areas at 400× magnification using integrated optical density (IOD) by Image Pro Plus 6.0 (Media Cybernetics, Inc., Silver Spring, MD, USA).

### 2.5. Multiplex Immunofluorescence Staining and Confocal Microscopy

Multiplex immunofluorescence staining for γδ TCR, STING and TBK1 was performed using 4 μm thick formalin-fixed, paraffin-embedded tissue sections. The experimental process was strictly in accordance with the protocol of the manufacturer of the multiplex immunohistochemistry/immunofluorescence staining kit (Absin, Shanghai, China, Catalog #abs50012). The primary antibodies anti-STING (1:200, Cell Signaling Technology, Catalog #D2P2F), anti-TBK1 (1:250, Abcam, Catalog #EP611Y) and anti-γδ TCR (1:10, Thermo Fisher, Catalog #5A6.E9) were used for incubation in sequence at room temperature for 60 min. TSA monochromatic fluorescent dyes 520, 570 and 650 were used in turn after incubation with dual anti-rabbit and mouse HRP-conjugated IgG at room temperature for 10 min. Antigen retrieval was performed at the beginning and the end of each round of staining. The sections were stained with DAPI and then were observed under a confocal laser microscope (Leica sp8, Wiesbaden, Germany), and images were overlaid in LAS X software.

### 2.6. Western Blotting Analysis

Fresh tissue samples from patients with OLP and healthy controls were carefully dissected. Protein concentrations were measured according to a BCA assay (Thermo Scientifc, Waltham, MA, USA) generated standard curve. A total of 40 µg of protein was denatured and then subjected to 12% SDS-polyacrylamide gel electrophoresis, followed by transfer onto polyvinylidene fuoride membranes (Millipore Corporation, Billerica, MA, USA). The membranes were incubated with primary antibodies (anti-γδ TCR (Thermo Fisher, Catalog #5A6.E9), anti-STING (Cell Signaling Technology, Catalog #D2P2F), anti-NAK/TBK1 (Abcam, Catalog #EP611Y), anti-p-STING (Cell Signaling Technology, Catalog #E9A9K), anti-p-TBK1 (Cell Signaling Technology, Catalog #D52C2), anti-IFN-γ (Abclonal, Wuhan, China, Catalog #A12450), anti-IL-17A (Bioworld, Dublin, OH, USA, Catalog #BS6041), anti-HLA-DR (Nonus, CO, USA, Catalog #NBP2-67610) and anti-CCR6 (Bioss, Beijing, China, Catalog #BS-1542R)). After being washed, the membranes were then probed with secondary antibodies. Next, the blots were stained using an enhanced chemiluminescence detection kit (Applygen, Beijing, China). The relative protein levels were calculated based on β-actin (Proteintech, Chicago, IL, USA, Catalog #66009-1-Ig) as the loading control and were densitometrically analyzed by Image J software (NIH, Bethesda, MD, USA).

### 2.7. Statistical Analysis

All statistical analysis was performed on Graph Pad Prism 8.0 (GraphPad Software Inc., La Jolla, CA, USA). Data are presented as mean ± SD for parametric tests and median (interquartile range) for non-parametric tests. Variance was determined by F test. Independent-samples *t-*tests (2 groups) or one-way analysis of variance (ANOVA) with Tukey’s multiple comparison test (3 or more groups) was used for data that were normally distributed and showed homogeneity of variance. Statistical analysis was performed using non-parametric Mann–Whitney U-tests (2 groups) or Kruskal–Wallis tests (3 or more groups) for non-normally distributed data. *p* < 0.05 was considered statistically significant.

## 3. Results

### 3.1. Aberrant Expression Pattern of γδ T Cells in Lesions and PBMCs of OLP Patients

A profound proportion of γδ T cells among CD3^+^ T cells was observed in OLP lesions when compared with healthy controls (OLP versus CON: 60.89% ± 13.78% versus 3.77% ± 0.651%, *p* = 0.0115; Figure 1A). The percentage of αβ T cells among CD3^+^ T cells was obviously increased in the lesions of OLP (OLP versus CON: 8.027% ± 1.777% versus 1.880% ± 1.344%, *p* = 0.0264; Figure 1B). Notably, the majority of CD3^+^ T cells in OLP lesions were of the γδ TCR lineage (Figure 1C). In addition, the γδ/αβ T cells ratio was significantly upregulated in OLP lesions (*p* = 0.0027; Figure 1D).

On the contrary, the peripheral counterparts of γδ T cells within the CD3 subset were obviously decreased in OLP patients (OLP versus CON: 2.285% ± 1.392% versus 6.56% ± 4.52%, *p* = 0.0045; Figure 1E). Specifically, both NEOLP and EOLP had a lower proportion of peripheral γδ T cells than that in healthy controls (EOLP versus CON: 2.11% ± 0.62% versus 6.56% ± 4.52%, *p* = 0.0005; NEOLP versus CON: 2.40% ± 1.75% versus 6.56% ± 4.52%, *p* = 0.0008; Figure 1F,G). However, no statistically significant difference was found in the percentage of peripheral γδ T cells between the clinical types of OLP.

### 3.2. Identification of Phenotypes of γδ T Cells in Lesions and PBMCs of OLP

We performed activation and antigen-presentation-associated phenotypic screening of γδ T cells and αβ T cells in tissue specimens from healthy volunteers and patients with OLP. Significantly higher percentages of CD69^+^ (*p* = 0.0492; Figure 2A) and NKG2D^+^ γδ T cells (*p* = 0.0072; Figure 2B) were found in OLP lesions. The proportions of HLA-DR^+^ (Figure 2C) and TLR4^+^ (Figure 2D) γδ T cells did not show any statistically significant differences between OLP and control groups. In addition, no statistically significant difference was found in the percentages of local CD69^+^, NKG2D^+^, HLA-DR^+^ and TLR4^+^ αβ T cells (Figure 2A–D). Notably, the proportions of CD69^+^ (*p* = 0.0362; Figure 2A), NKG2D^+^ (*p* = 0.0045; Figure 2B) and HLA-DR^+^ γδ T cells (*p* = 0.0075; Figure 2C) were obviously elevated compared to the same phenotypes of local αβ T cells from OLP patients. Furthermore, the pro-inflammation cytokine IFN-γ was strongly produced by OLP γδ T cells (*p* = 0.0059; Figure 2E).

Unlike their local counterparts, peripheral HLA-DR^+^ (OLP versus CON: 40.41% ± 10.20% versus 29.48% ± 11.24%, *p* = 0.025; Figure 3A) and TLR4^+^ γδ T cells (*p* = 0.049; Figure 3B) were significantly increased in number in OLP, whereas NKG2D^+^ γδ T cells (OLP versus CON: 59.48% ± 22.52% versus 75.89% ± 13.28%, *p* = 0.038; Figure 3C) obviously decreased.

### 3.3. Upregulated γδ T Cells and STING-TBK1 Pathway in OLP Lesions

As shown in Figure 4, obvious γδ TCR^+^ cells were detected in OLP lesions (OLP versus controls: 0.32 ± 0.10 versus 0, *p* = 0.0132; Figure 4A1–A5). These γδ T cells were mainly located on cell membranes in the lymphocytic infiltration area adjacent to the basal membrane of the epithelium.

The expression levels of STING (OLP versus controls: 0.50 ± 0.06, *p* = 0.0394; Figure 4B1–B5) and TBK1 (OLP versus controls: 0.48 ± 0.04, *p* = 0.0022; Figure 4C1–C5) proteins were significantly upregulated in OLP. Both STING and TBK1 were mostly accumulated in the cytoplasm in the basement membrane and in the lymphocytic infiltrate of the lamina propria.

### 3.4. Co-Localization of STING^+^TBK1^+^ in γδ T Cells of OLP Lesions

Multiplex immunofluorescence staining assays were utilized to verify the involvement of STING-TBK1 pathway in γδ T cells. As shown in Figure 5, obvious STING (Green-labelled), TBK1 (Red-labelled) and γδ TCR (Pink-labelled) co-localization was observed in lesions of OLP. These STING^+^TBK1^+^ γδ T cells were mainly distributed in the lamina propria of OLP lesions, suggesting the existence of interaction between the STING-TBK1 pathway and γδ T cells in OLP lesions.

### 3.5. The Regulatory Mechanisms of γδ T Cells in OLP Lesions

In accordance with aforementioned results, Western blotting analysis displayed that γδ T cells and the STING-TBK1 pathway were significantly upregulated in OLP lesions (Figure 6). Specifically, the lesions of EOLP showed increased expression of γδ TCR (*p* < 0.0001; Figure 6A) and IL-17A (*p* < 0.0001; Figure 6B). In addition, STING (*p* = 0.0101) and TBK1 (*p* < 0.0001) proteins, and p-STING (*p* = 0.0014) and p-TBK1 (*p* = 0.0021), were significantly increased in lesions of OLP (Figure 6C,D). Moreover, the expression levels of pro-inflammation cytokines IL-17A (*p* < 0.0001; Figure 6B) and IFN-γ (*p* = 0.0019; Figure 6C) were significantly elevated in OLP lesions, and the same tendencies of the γδ T cells and STING-TBK1 pathway were present. Both the γδ TCR and IL-17A protein expression showed a significant difference between the two types of OLP (EOLP > NEOLP: *p* = 0.0427 and *p* < 0.0001, respectively; Figure 6A,B). In addition, OLP lesions expressed more chemokine receptor CCR6 protein (*p* = 0.0009; Figure 6E) and HLA-DR (*p* < 0.0001; Figure 6F).

## 4. Discussion and Conclusions

γδ T cells connect innate and adaptive immunity and play regulatory roles in many autoimmune diseases [28]. Examination of the common inflammatory skin disease psoriasis revealed obvious upregulation of γδ T cells in psoriatic skin and a reduction in γδ T cells in peripheral blood [11]. In the present study, overexpression of γδ T cells was found in OLP lesions, whereas peripheral OLP γδ T cells were remarkably decreased in number. Moreover, HLA-DR was upregulated in peripheral γδ T cells and CD69 was elevated on local γδ T cells, suggesting the activation of γδ T cells in OLP. We recently identified that the cell-motility-associated protein CD103 was significantly expressed on peripheral γδ T cells of OLP lesions [29]. In addition, the present data revealed upregulated expression of CCR6, accompanied by increased levels of γδ T cells and HLA-DR in OLP-lesion tissues. CCR6, expressed on skin-resident γδ T cells, can interact with its sole ligand CCL20—which is expressed by keratinocytes—and is vital for recruitment of activated γδ T cells to skin [30]. It is speculated that activated γδ T cells exhibited a tissue tropism from peripheral blood to a local lesional mucosa compartment of OLP.

Unlike αβ T cells, γδ T cells can act in different physiological contexts by harnessing TLRs and NKG2D, which further substantially induce pro-inflammatory cytokines and chemokines’ production upon activation [31,32,33,34,35]. TLRs have been proven to nurture the functions of γδ T cells directly [36], which are especially essential for the secretion of IFN. Both TLR3 and TLR4 ligands promoted the IFN-γ production by γδ T cells in a type-I-IFN-dependent manner [37]. NKG2D has been recognized as a co-effector for the TCR activation of γδ T cells [38]. In the present study, we found upregulated frequencies of TLR4 and NKG2D on peripheral and local γδ T cells of OLP, respectively. Notably, the proportions of CD69^+^, HLA-DR^+^ and NKG2D^+^ γδ T cells were obviously elevated compared to the same phenotypes of local αβ T cells in OLP. IFN-γ produced by local γδ T cells, but not local αβ T cells, was significantly enriched in OLP lesions. Therefore, infiltrating γδ T cells may be more energetic and function as the main contributor to the immunoregulatory responses in OLP lesions.

A reduction in peripheral γδ T cells was negatively correlated with disease severity of psoriasis, SLE and multiple sclerosis, suggesting an association of γδ T cells with disease severity [11,12,39]. Our recent study demonstrated a positive correlation between γδ T cells’ expression and disease severity (RAE scores) of OLP [29]. The present data showed that γδ T cells strikingly accumulated in the lesional mucosa of a severe clinical type of OLP (EOLP). Accordantly, the expression levels of pro-inflammation cytokines IL-17A and IFN-γ were significantly elevated in OLP lesions. There was especially more IL-17A protein in the lesional mucosa of EOLP. Activated γδ T cells could amplify the Th17 response and induce αβ T cell immune responses, which aggravate graft-versus-host disease [40]. Psoriatic γδ T cells expressed IL-17A and activated keratinocytes in a IFN-γ and TNF-α-dependent manner [11]. Hence, the pro-inflammatory crosstalk between γδ T cells and keratinocytes may participate in the development of OLP.

The STING-TBK1 pathway detects pathogenic or endogenous DNA to trigger an innate immune reaction, initiating a strong type I IFN response, and serves as an important pathway in autoimmunity [36,41]. The activation of the STING-TBK1 pathway could promote the secretion of IFN-γ by T cells [25,26]. Our previous studies, and others, have described that IFN-γ significantly infiltrates in local tissues of OLP and regulates the apoptotic process of keratinocytes [18,42,43,44]. Recently, the activation of the STING-TBK1 pathway has been found to promote cytokine induction in short-term-expanded γδ T cells [27]. STING knockdown inhibited the OLP-derived, cell-free, DNA-induced inflammation in THP-1 macrophages and inflammatory responses [45]. According to the present data, STING^+^TBK1^+^ γδ T cells were mainly present in the lamina propria of OLP lesions. Notably, STING and TBK1 protein levels, and p-STING and p-TBK1 proteins were elevated in the lesional mucosa of OLP, in accordance with the expression of γδ T cells and IFN-γ. The STING-TBK1 pathway was considered to be involved in pro-apoptotic signaling and to contribute to cell death [26]. Moreover, γδ T cells could exert cytotoxic activity under certain stressed circumstances [46]. The activated STING-TBK1 pathway in γδ T cells might regulate the apoptosis of basal keratinocytes, which is identified to be of great importance for the development of OLP. In addition, STING primordially activates autophagy in a TBK1 and IFN-dependent manner [47]. Our recent study indicated that IFN-γ could activate IRGM-mediated autophagy in T cells and decrease the proliferation and apoptosis of T cells cocultured with keratinocytes, which might participate in the immunoregulatory mechanism of OLP [48]. Collectively, these results suggest that the upregulated STING-TBK1 pathway might be complementary to activated γδ T cells during the IFN-mediated immune responses of OLP.

In conclusion, these findings indicate that aberrant activation of the STING-TBK1 pathway in γδ T cells may regulate the immune response in OLP, probably via pro-inflammatory cross-talk between γδ T cells and keratinocytes. Functional studies of isolated γδ T cell subtypes and further in situ studies are needed, and the manipulation of γδ T cells and the STING-TBK1 pathway might serve as a promising target for redefining the immunoregulatory mechanism of OLP.

## Figures and Tables

**Figure 1 biomedicines-11-00955-f001:**
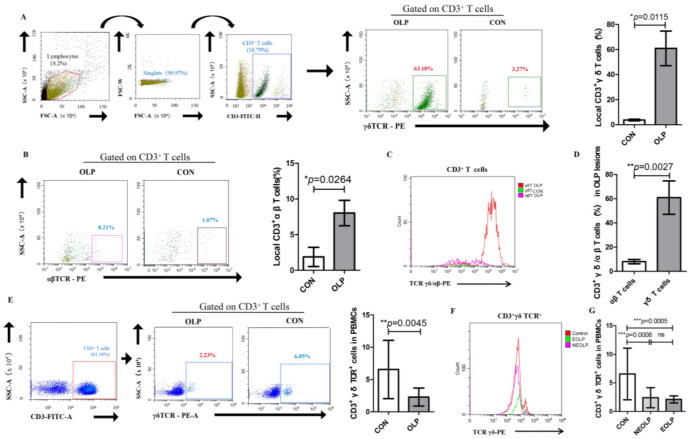
Altered frequencies of local and peripheral γδ T cells in OLP. (**A**) The expression of CD3^+^ γδ TCR^+^ cells in OLP lesions was significantly increased. (**B**) The upregulated proportion of CD3^+^αβ TCR^+^ cells in OLP lesions. (**C**) The histogram of subsets of CD3^+^ T cells in OLP lesions and healthy controls. (**D**) The local γδ/αβ T cells ratio obviously increased in OLP. (**E**) The frequency of peripheral CD3^+^ γδ TCR^+^ cells was significantly decreased in OLP. (**F**,**G**) The expression patterns of peripheral γδ T cells in different clinical types of OLP (erosive OLP (EOLP); nonerosive OLP (NEOLP)) and healthy controls (CON). Data are presented as mean ± SD. Statistical analysis was performed using independent-samples *t-*tests (2 groups) or one-way analysis of variance (ANOVA) with Tukey’s multiple comparison test (3 or more groups). * *p* < 0.05, ** *p* < 0.01, *** *p* < 0.001, ns: non-significant, *p* > 0.05. CON, healthy controls; OLP, oral lichen planus; NEOLP, non-erosive oral lichen planus; EOLP, erosive oral lichen planus.

**Figure 2 biomedicines-11-00955-f002:**
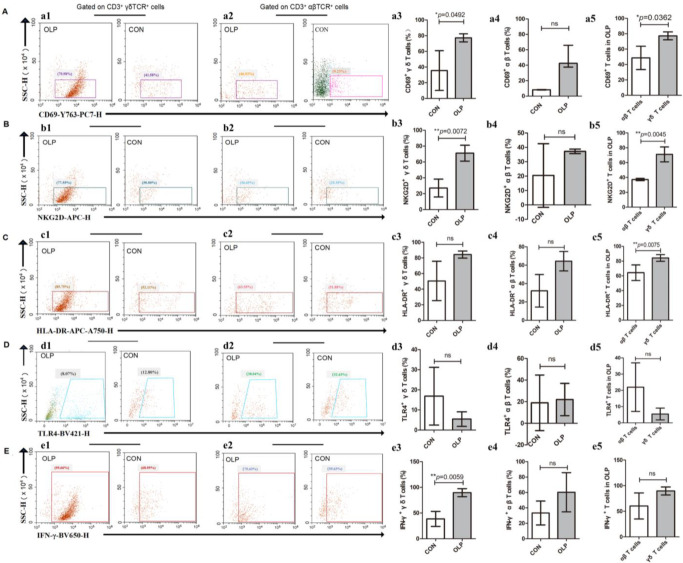
The expression profiles of CD69, NKG2D, HLA-DR, TLR4 and IFN-γ in local CD3^+^ T cell subsets. (**A**) Representative plots of CD69^+^ CD3^+^ γδ TCR^+^ cells and CD69^+^ CD3^+^ αβ TCR^+^ cells. (**B**) Representative plots of NKG2D^+^ CD3^+^ γδ TCR^+^ cells and NKG2D^+^ CD3^+^ αβ TCR^+^ cells. (**C**) Representative plots of HLA-DR^+^ CD3^+^ γδ TCR^+^ cells and HLA-DR^+^ CD3^+^ αβ TCR^+^ cells. (**D**) Representative plots of TLR4^+^ CD3^+^ γδ TCR^+^ cells and TLR4^+^ CD3^+^ αβ TCR^+^ cells. (**E**) Representative plots of IFN-γ^+^ CD3^+^ γδ TCR^+^ cells and IFN-γ^+^ CD3^+^ αβ TCR^+^ cells. Data are presented as mean ± SD, and we applied independent-samples *t*-tests (* *p* < 0.05, ** *p* < 0.01, ns: non-significant, *p* > 0.05), except for a4 and d4, for which the data are presented as median (interquartile range). Statistical analysis was performed using non-parametric Mann–Whitney U-tests for a4 and d4 (ns: non-significant, *p* > 0.05. CON, healthy controls; OLP, oral lichen planus).

**Figure 3 biomedicines-11-00955-f003:**
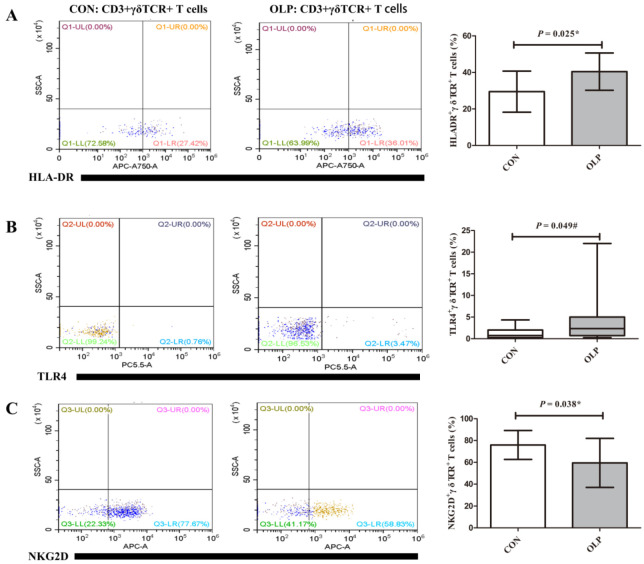
Aberrant expression of activation and antigen-presentation-associated molecules in the γδ T cells. (**A**) The expression of HLA-DR was significantly enhanced in the gated γδ T cells from peripheral blood mononuclear cells (PBMCs) of OLP. Data are presented as mean ± SD. Statistical analysis was performed using independent-samples *t-*tests. * *p* < 0.05. (**B**) The proportion of TLR4^+^ γδ T cells was significantly elevated in OLP PBMCs. Data are presented as median (interquartile range). Statistical analysis was performed using non-parametric Mann–Whitney U-tests. # *p* < 0.05. (**C**) The circulating NKG2D^+^ γδ T cells were significantly decreased in OLP PBMCs. Data are presented as mean ± SD. Statistical analysis was performed using independent-samples *t-*tests. * *p* < 0.05. CON, healthy controls; OLP, oral lichen planus.

**Figure 4 biomedicines-11-00955-f004:**
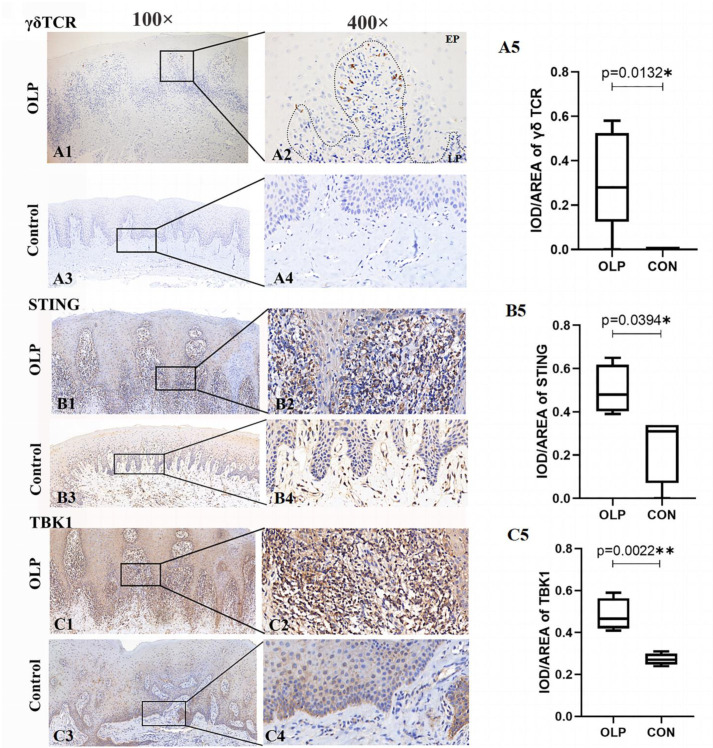
The increased expression levels of γδ TCR, STING and TBK1 in lesion tissues of OLP. The expression of γδ TCR in OLP-lesion tissues (**A1**,**A2**) and healthy controls (**A3**,**A4**). The expression of STING in OLP-lesion tissues (**B1**,**B2**) and controls (**B3**,**B4**). The frequency of TBK1 (**C1**,**C2**) in OLP lesions and controls (**C3**,**C4**). The results were assessed by IOD/area values (**A5**,**B5**,**C5**). Magnification: (**A1**,**A3**,**B1**,**B3**,**C1**,**C3**): 100×; (**A2**,**A4**,**B2**,**B4**,**C2**,**C4**): 400×. Data are presented as mean ± SD. Statistical analysis was performed using independent-samples *t-*tests. * *p* < 0.05, ** *p* < 0.01. CON: healthy controls; OLP: oral lichen planus; EP: epithelium; LP: lamina propria; IOD: integrated optical density.

**Figure 5 biomedicines-11-00955-f005:**
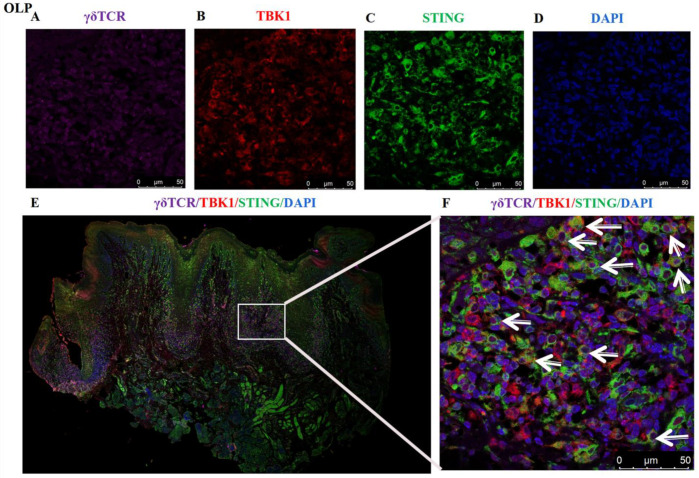
Co-localization of STING and TBK1 in γδ T cells of OLP lesions. (**A**) pink: γδ TCR-650 nm; (**B**) red: TBK1-RFP 488nm; (**C**) green: STING-GFP 520nm; (**D**) blue: nuclear-DAPI (magnification: 630×). (**E**) The entire field of immunofluorescent image (magnification: 10×), STING^+^ TBK1^+^ γδ T cells mainly exist in the lymphocytic infiltration area of lamina propria in OLP lesions. (**F**) White arrows represent for these γδ TCR^+^ cells positively expressing STING and TBK1 (magnification: 630×). The slides were observed under a confocal laser microscope (Leica sp8, Germany), and images were overlaid in LAS X software.

**Figure 6 biomedicines-11-00955-f006:**
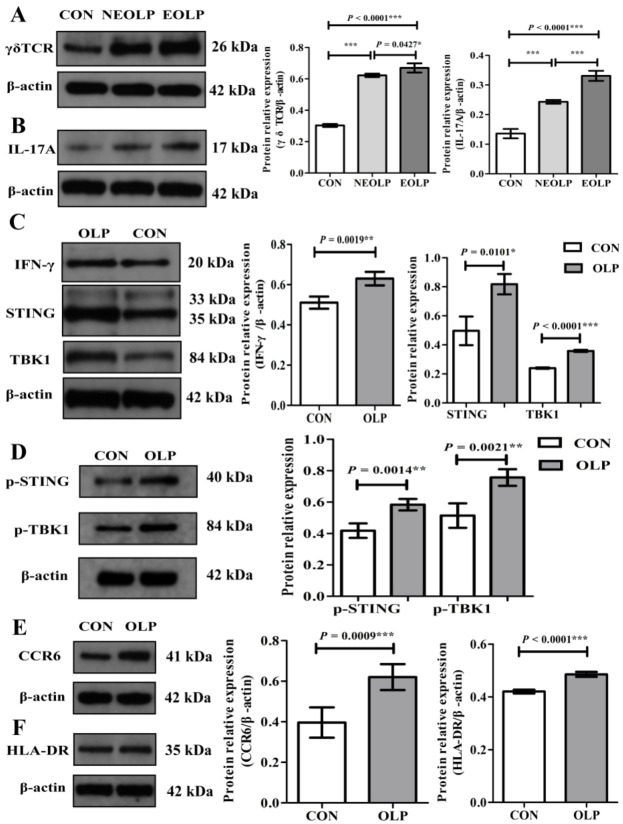
The activation of γδ T cells and the STING-TBK1 pathway in OLP lesions and controls. (**A**,**B**) The expression of γδ TCR and IL-17A proteins was upregulated in OLP, especially in lesions of erosive OLP. (**C**,**D**) Increased IFN-γ protein and activation of the STING-TBK1 pathway in OLP lesions. (**E**,**F**) The frequencies of CCR6 and HLA-DR proteins were elevated in OLP lesions. Data are presented as mean ± SD. Statistical analysis was performed using independent-samples *t-*tests (CON versus OLP) or one-way analysis of variance (ANOVA) with Tukey’s multiple comparison test (CON versus EOLP versus NEOLP). * *p* < 0.05, ** *p* < 0.01, *** *p* < 0.001. CON, healthy controls (n = 3); OLP, oral lichen planus (n = 3); NEOLP, non-erosive oral lichen planus (n = 2); EOLP, erosive oral lichen planus (n = 2); STING, stimulator of interferon genes; TBK1, TANK-binding kinase 1; p-STING, phosphorylated-stimulator of interferon genes; p-TBK1, phosphorylated-TANK-binding kinase 1.

**Table 1 biomedicines-11-00955-t001:** Clinical features of the subjects.

	OLP (*n* = 39)	Control (*n* = 27)
**Gender**		
Male	18	13
Female	21	14
**Ages (y)**		
Range	23~69	21~61
Mean ± SD	44.21 ± 2.079	34.74 ± 3.205
**Clinical Forms**		
Erosive	17	
Nonerosive	22	
**RAE**		
Range	1~16	
Mean ± SEM	6.53 ± 3.59	

## Data Availability

The data that support the finding of this study are available from the corresponding author Zhou Gang, upon request.

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
