# Peer review of "Aberrant Activation of the STING-TBK1 Pathway in γδ T Cells Regulates Immune Responses in Oral Lichen Planus"

_biomedicines, 2023, doi:10.3390/biomedicines11030955_

Round 1

Reviewer 1 Report

The authors aimed to study, the expression pattern of γδ T cells in the peripheral and local immune environment of OLP. Meanwhile, activation and antigen presentation associated molecules were detected on γδ T cells. The expression level of IFN-γ secreted by γδ T cells were also measured. Besides, the activation of STING-TBK1 pathway, together with their co-localization with γδ T cells were explored to identify the possible involvement of γδ T cells and STING-TBK1 pathway in the pathogenesis of OLP.

The study covers some issues that have been overlooked in other similar topics. The structure of the manuscript appears adequate and well divided in the sections. Moreover, the study is easy to follow, but some issues should be improved. Some of the comments that would improve the overall quality of the study are:

I-) Authors must pay attention to the technical terms acronyms they used in the text;

II-) Please better stated the limitation of the study.

Author Response

Point 1: Authors must pay attention to the technical terms acronyms they used in the text.

Response 1:

    We appreciate your comments! 

    These technical terms acronyms used in the text have revised and double-checked in the revised manuscript. The revised parts have been marked in red in our revised manuscript.

Point 2: Please better stated the limitation of the study.

Response 2:

     Thank you for your suggestions! 

    The present study aims to explore the involvement of γδ T cells and their potential mechanism in the pathogenesis of OLP. We first compared the discrepancy between local and peripheral γδ T cells, and detected the activation of STING and TBK1 in γδ T cells of OLP. The results showed that γδ T cells were significantly upregulated in the lesional tissues, whereas their peripheral counterparts decreased in OLP. Specifically, these γδ T cells exhibited activated profiles. Furthermore, significant co-localization of STING and TBK1 were observed in γδ T cells of OLP lesions. In addition, enhanced IFN-γ and IL-17A were positively associated with activated STING-TBK1 pathway and γδ T cells in OLP.

    Our recent study revealed that the percentage of peripheral γδ T cells  decreased in OLP [1]. The proportion of CD103+, PD-1+ and PD-L1+ γδ T cells was upregulated, whereas IL-4+, granzyme B+ and TNF-α+ γδ T cells were reduced in OLP peripheral blood [1]. Additionally, present study demonstrated increased frequencies of local CD69+ and NKG2D+ γδ T cells and peripheral HLA-DR+ and TLR4+ γδ T cells in OLP. Therefore, the previous and present studies mainly focus on the functional parameters of whole γδ T cells in OLP.

     In humans, γδ T cells are classified according to their Vδ gene segment used. Until now only 3 true Vδ genes (Vδ1-3) exist; and 7 functional Vγ gene segments (Vγ2-5, Vγ8, Vγ9, and Vγ11)[2]. Vδ1+ T cells are abundant in the human epithelium, whereas Vδ2+ T cells are the main subset present in peripheral blood and this δ chain is generally associated to the Vγ9 [2]. For further study, we will continue to explore these isolated γδ T cell subtypes at the molecular and functional level in the pathogenesis of OLP.

References

  1. Yang JY, Wang F and Zhou G. Characterization and function of circulating mucosal-associated invariant T cells and γδT cells in oral lichen planus, J Oral Pathol Med 2021;51:74-85.
  1. Shiromizu CM andJancic CC.γδ T Lymphocytes: An Effector Cell in Autoimmunity and       Infection, Front Immunol 2018;9:2389-2389.

Reviewer 2 Report

Title  OK

Introduction: The introduction  is OK

Material and methods are well developed the methods are used appropriate

You should specify more clearly indicate the criteria for inclusion Oral lichen planus / oral lichenoid disease. Do I include dysplasia?

Was the tissue biopsy and blood sample taken on the same day?

The results are supported by tables  that improve the interpretation of the results. Some of the results sections are repetitive

It should highlight the strong points and add the limitations of the study  

might serve as a promising target for redefining the immunoregulatory mechanism of OLP.

 References are up to date

Reviewer 3 Report

To Authors 

- Title

The title communicates distinctly what the manuscript is about, identifying the report as an article on the topic of immune response in oral lichen planus; no unnecessary description reported.  

- Abstract 

The abstract provides an explicit statement of the main objectives the article addresses; it specifies the protocol methods applied and the techniques used for the research.

-  Introduction

The introduction describes the rationale for the research in the context of what is already known; the explicit statement of the research questions and aims the article address are clearly expressed in the introduction.

-  Materials and methods 

The section is correctly and clearly divided into sub-paragraphs each addressing the various methods used for the corresponding questions and objectives in relation to the activation of STING-TBK1 pathway in gd T cells responses. 

-  Discussion and conclusion 

In the discussion section, the findings of the research are logically explained by the main topics. Given the complexity of the discussion, it would be appropriate to insert a final paragraph of conclusions (although not mandatory). In this section, the implications of the findings for future research and potential applications are considered, but they should be further developed in a conclusive section.

Author Response

Point 1: - Title

The title communicates distinctly what the manuscript is about, identifying the report as an article on the topic of immune response in oral lichen planus; no unnecessary description reported.  

Response 1:

     Thank you for your appreciation.

Point 2: - Abstract 

The abstract provides an explicit statement of the main objectives the article addresses; it specifies the protocol methods applied and the techniques used for the research.

Response 2:

     Thanks for your appreciation.

Point 3: -  Introduction

 The introduction describes the rationale for the research in the context of what is already known; the explicit statement of the research questions and aims the article address are clearly expressed in the introduction.

Response 3:

     Thanks a lot!

Point 4: -  Materials and methods 

The section is correctly and clearly divided into sub-paragraphs each addressing the various methods used for the corresponding questions and objectives in relation to the activation of STING-TBK1 pathway in gd T cells responses. 

Response 4:

    Thank you for your appreciation.

Point 5: -  Discussion and conclusion 

In the discussion section, the findings of the research are logically explained by the main topics. Given the complexity of the discussion, it would be appropriate to insert a final paragraph of conclusions (although not mandatory). In this section, the implications of the findings for future research and potential applications are considered, but they should be further developed in a conclusive section.

Response 5:

    We appreciate your suggestions! 

    A final paragraph of Conclusions has been revised in the Discussion and Conclusion section of our revised manuscript as follows.

    In conclusion, these findings indicated that aberrant activation of STING-TBK1 pathway in γδ T cells may regulate immune response in OLP, probably via pro-inflammatory cross talk between γδ T cells and keratinocytes. Functional studies of isolated γδ T cell subtypes and further in situ studies are needed, and the manipulation of γδ T cells and STING-TBK1 pathway might serve as a promising target for redefining the immunoregulatory mechanism of OLP.